# Machine Learning Methods for Automatic Silent Speech Recognition Using a Wearable Graphene Strain Gauge Sensor

**DOI:** 10.3390/s22010299

**Published:** 2021-12-31

**Authors:** Dafydd Ravenscroft, Ioannis Prattis, Tharun Kandukuri, Yarjan Abdul Samad, Giorgio Mallia, Luigi G. Occhipinti

**Affiliations:** Department of Electrical Engineering, University of Cambridge, Cambridge CB3 0FA, UK; ip355@cam.ac.uk (I.P.); trk25@cam.ac.uk (T.K.); yy418@cam.ac.uk (Y.A.S.); gm662@cam.ac.uk (G.M.); lgo23@cam.ac.uk (L.G.O.)

**Keywords:** artificial neural networks, graphene, machine learning, silent speech recognition, strain gauge

## Abstract

Silent speech recognition is the ability to recognise intended speech without audio information. Useful applications can be found in situations where sound waves are not produced or cannot be heard. Examples include speakers with physical voice impairments or environments in which audio transference is not reliable or secure. Developing a device which can detect non-auditory signals and map them to intended phonation could be used to develop a device to assist in such situations. In this work, we propose a graphene-based strain gauge sensor which can be worn on the throat and detect small muscle movements and vibrations. Machine learning algorithms then decode the non-audio signals and create a prediction on intended speech. The proposed strain gauge sensor is highly wearable, utilising graphene’s unique and beneficial properties including strength, flexibility and high conductivity. A highly flexible and wearable sensor able to pick up small throat movements is fabricated by screen printing graphene onto lycra fabric. A framework for interpreting this information is proposed which explores the use of several machine learning techniques to predict intended words from the signals. A dataset of 15 unique words and four movements, each with 20 repetitions, was developed and used for the training of the machine learning algorithms. The results demonstrate the ability for such sensors to be able to predict spoken words. We produced a word accuracy rate of 55% on the word dataset and 85% on the movements dataset. This work demonstrates a proof-of-concept for the viability of combining a highly wearable graphene strain gauge and machine leaning methods to automate silent speech recognition.

## 1. Introduction

According to the WHO, around 5% of the population worldwide have hearing and speech impairments [1]. Silent communication is a technique that can help people with these conditions speak properly due to the conversion of silent attempts to speak into speech. This approach is significantly important for patients who cannot rely on traditional voice signals. For instance, it can help individuals who have undergone laryngectomies and may require speech training after surgery to speak clearly and confidently. More than 175,000 cases of laryngeal cancer had been reported only in 2018 [2]. Aside from helping individuals with speech impairments, this technology could also be used in areas where reliable and secure sound delivery is required, such as in locations of high noise.

Over the recent years, various approaches have been made in the field of silent speech recognition. There has been increasing success in the field with visual speech recognition [3,4,5] and EMG [6,7]. However, due to wearability difficulties, these solutions are not considered feasible for real-time communications. However, the growing popularity of flexible electronics has made them a promising alternative for silent communications. Flexible electronics is a promising solution for silent communications due to their reliability, stability, comfort and convenience.

Graphene strain sensors are used for various health monitoring applications. They can be used for monitoring physiological processes such as blood pressure, pulse rate, muscle movement, and gesture recognition [8,9]. A strain gauge is a type of sensor that measures the resistance of a device when it is subjected to strain, and it can be used to measure small movements. There are several factors required for attaching a strain gauge to the human body. Sensing vocal activity requires highly sensitive sensors that have a high gauge factor (GF)—a measure of how much the resistance changes with an applied force. Higher levels make strains more detectable, and also produce clearer signals. Graphene has been known to have a high gauge factor and this is a promising choice for sensors. Aside from having a high GF, other factors such as durability and flexibility are also key to long-standing performance. Xu et al. presented a strain sensor that is flexible and long-term-wearing. They used reduced graphene oxide (rGO)/deionised water sensing liquids and Ecoflex to produce the sensor [10]. Highly stretchable and long lasting strain sensor was able to be produced, due to the geometric structures and the connection types of the sensing materials. Similarly, Liu et al. use Ecoflex as an encapsulant surrounding a graphene/glycerol/potassium chloride ionic conductor as the sensing element [11]. They show that attaching a sensor to the throat can improve the quality of vocalisation, although this method has limited use. Wei et al. proposed a graphene-based wearable device that converts the vibrational cord and motion of the larynx into an electrical signal, which can be interpreted as a sound [12]. Wan et al. focused on developing a strain sensor that could pick up subtle vibrations of the vocal cords and radial artery pulses, with high conformity to the skin, and maximised contact area [13]. Finally, a wearable strain sensor made of graphene has been demonstrated by dying a polymer fabric with rGO, which can be used to detect the movements of humans when is applied in clothing [14]. It has high stability and comfort, and demonstrates an ability to detect a range of subtle human motions including vocal vibration.

These studies mainly show the sensor’s ability to identify the difference between a set of phonations and the repeatability between them. However, they do not attempt to classify and/or quantify the signals. In this paper we propose an approach which allows the classification of recorded resistance signals into predicted words. Machine learning approaches have been widely demonstrated for a range of supervised classification problems involving temporal data including video recognition, language translation, stock prediction, phoneme identification, quality of speech analysis [15]. The most similar areas to our strain gauge readings are those involving the classification of audio, vibrations and EEG signals. Many machine learning approaches perform classification based on the input of pertinent features extracted form the raw data. Mel-frequency cepstrum coefficient and time-domain features such as root mean square, variance and skewness have commonly been used for such applications [16,17,18]. The k-nearest neighbour classifier has been demonstrated in speech classification and speech emotion recognition [19,20]. Random forests have also been demonstrated for audio based problems including lip reading, speech emotion classification and audio signal classification [21,22,23]. Artificial neural networks (NNs) have revolutionised machine learning in the last few years, producing state-of-the-art performance in a wide range of applications including image classification [24], image segmentation [25], medical imaging [26] and temporal problems such as language translation [27], raw audio generation [28], visual speech recognition [4]. NNs combine feature representation learning and supervised discrimination into a uniform end-to-end training framework removing the need for preliminary feature extraction. In particular, convolutional-recurrent architectures have been shown to perform state-of-the-art results in a range of temporal problems. By utilising these machine learning approaches we aim to be able to make predictions on intended speech based on the readings of the throat-worn strain gauge sensor.

In this paper we address the challenge of using a strain gauge to predict speech, developing on our previous work [29]. The novelties of our work include: (1) Instead of just dealing with a small selection of words or throat movements, this study has a more robust set of words and a more detailed dataset. We designed a dataset with a wider selection of words which are specifically chosen and with multiple repetitions of each word. (2) Thisstudy proposes a silent speech recognition pipeline able to automatically recognise and categorise words. Other studies do not provide any algorithmic method for the automated classification of words with results demonstrated in a visual manner. We developed and implemented a selection of machine learning algorithms—NNs, random forests and k-nearest neighbour—to classify input signals. (3) In order to quantify the performance of sensors, we introduce testing of our word classifiers to provide a measurable output to show the accuracy of the combined sensor and classification system. This paper develops on our previous work by exploring multiple machine learning approaches including random forests and k-nearest neighbour classifiers utilising handcrafted feature extraction methods. This is in addition to the combined feature learning and classification pipeline of neural networks. Additionally, we have developed an expanded dataset which has led to higher accuracies for both the words and movements datasets.

## 2. Materials and Methods

The purpose of this work is to create an automated method for the recognition of silent speech. To achieve this it is necessary to be able to detect a signal related to speech and be able to classify what is being said. To achieve this we break our proposed approach into three steps as shown in Figure 1: (1) Device Fabrication—A device is required to be able to detect signals related to speech. We develop a graphene strain gauge sensor to be worn on the throat for this purpose. (2) Data Generation—In order to be able to learn the connection between the obtained signals and the speech they represent it is necessary to collect a suitably sized dataset. (3) Machine Learning Classification—Multiple machine learning techniques are used to analyse the relationship between acquired signal and intended speech, with the aim of developing an algorithm which can make an accurate mapping between the two and make correct predictions of speech from previously unseen signals.

### 2.1. Device Fabrication

The purpose of the sensor is to be able to detect non-audio signals related to speech. The key requirements of such a device are that it be non-invasive and produce a clear measurable signal response. As such, we propose to develop a small, flexible, wearable strain gauge sensor which can detect small movements and vibrations of the throat. There are a number of important characteristics which are required to produce a suitable sensor for our purpose. Strain gauges measure the amount of deformation of an object. Key properties of such devices include sensitivity, fast response times, stretchability, stability and durability. Graphene is a useful material for flexible devices due to its range of advantageous properties including high thermal and electrical conductivity, large specific surface area and high mechanical flexibility [30]. A large specific surface area is important for use as an external health monitoring sensor as it allows a high level of sensitivity. A high mechanical flexibility and relative thinness makes the device highly wearable by allowing good conformity with the skin.

We propose to use graphene deposited on lycra fabric as described previously in [29]. This methodology offers high flexibility and wearability. The authors of [14] have demonstrated the applicability of graphene as a colourant on textile fabric to produce a strain sensor. This approach, unlike many others, does not require polymer encapsulation. Polymer encapsulation has the disadvantages of poor fit to the human body, poor wearability, low comfort and a complex fabrication process. Fabricating directly onto lycra textile without any polymer encapsulation allows for a more wearable and close-fitting sensor.

In Figure 1 we illustrate the steps involved in the fabrication process. The method consisted of screen printing a layer of graphene ink onto fabric. The graphene ink was produced via liquid phase exfoliation using the method proposed by Marcano [31]. This produces flakes in the range 50–150 nm with thicknesses of 1–3 nm. The fabric used was lycra fabric as it has good flexibility, stretchability and wearability. Screen printing was used to transfer the graphene ink to the lycra fabric with the steps of printing and drying repeated until the graphene fully covered the desired section of fabric. The fabric was then cut down to a size—our throat sensor measured 35 mm × 25 mm with copper electrodes being attached using silver paste. The device was then allowed to dry, being placed in an oven helped this drying process. This is simple, cheap and fast process which produces a flexible, close-fitting and wearable sensor exploiting the high useful properties of graphene.

### 2.2. Data Generation

The development of a suitable dataset is required in order to train classification algorithms to produce quantitative results for the accuracy of our silent communication system. We developed two distinct datasets: one for words and one for noises/head movements. Each had enough repetitions so as to be sufficient for classification. The list of words was selected from the w-22 phonetically balanced word set [32]. Using a phonetically balanced word set allows for the differentiation of commonly used sounds within the English language. A set of 15 words were chosen: as, dad, felt, give, hum, it, low, me, none, or, poor, there, true, up, us. A selection of basic noises/movements was also tested. In addition to words, a dataset from recordings of coughing, swallowing, yawning and nodding was also made. The movements for the members of this dataset are larger and more distinct than for words which should make classification of the resultant signals easier. Multiple repetitions are necessary to find the features which discriminate between different words but are consistent across the same word. Each noise was repeated 20 times by the speaker.

The graphene strain gauge sensor was attached to the subject’s throat as shown in Figure 1. The copper electrodes were attached to an ohmmeter to allow resistance readings to be recorded. Each of the words and movements were measured for 20 repetitions in turn with a suitable gap between each. The data was afterwards labelled and timings recorded. The recordings were split into equal sized windows of 3000 ms each centred on the phonation with no overlap between phonations. The equal sized windows are necessary as neural networks require all inputs to be equally sized. All recordings were made on one subject during one sitting to ensure that sensor position remained constant.

### 2.3. Classification

Our approach can be seen as a classified supervised problem. We want to understand the mapping between measured resistance changes from the strain gauge and the intended speech. To achieve this we want to utilise machine learning techniques which learn from a training set how to discriminate between inputs allowing them to make accurate predictions on unseen samples. We propose two ways of doing so (1) Using handcrafted feature extraction methods to obtain the pertinent aspects of the signals, from which machine learning algorithms, including random forests and k-nearest neighbour, can be trained (2) Using neural networks to automatically learn the important features and perform classification on test signals.

#### 2.3.1. Feature Extraction

The choice of features is a non-trivial problem and is subjective depending on what you expect to be important within your data. Each feature extraction technique is useful for a certain purpose and none are ubiquitously good across all problems. As the signal we are measuring, resistance changes caused by small movements in the throat, are unique there is not a pre-existing standard for the pertinent features for describing the signal. In this paper, we explore a number of feature extraction techniques which have been demonstrated in approaches which use similar signals. In particular, we consider feature extraction methods used for audio, vibrations and EEG signals [16,17]. We consider time-domain features including the root mean square, variance, skewness, kurtosis, shape factor and entropy. We also calculate the Mel-frequency cepstrum coefficient. This is an audio processing technique which is used as a compact representation of a audio signal’s power spectrum and are widely used in speech recognition [18].

The features are combined into a single array which are then used by the machine learning algorithms for training or classification. This approach decreases the size of the input to the machine learning algorithms, allowing them to distinguish between the important features without overfitting to the training set.

#### 2.3.2. Machine Learning Algorithms

From the set of extracted features we want to be able to make a map between these and their associated noise, making this a supervised classification problem. Machine learning models are useful for this kind of problem as they are able to learn the relationship between the input features and the output category. They are fed the set of training data feature vectors and corresponding categories. Using these they adjust internal methods to create a specific model suited to correctly classify this data. The model is then given the testing data feature vectors and the produced output is compared to the vector’s actual category. Certain parameters for the models can be changed and the models retrained and retested to incrementally improve the accuracy of the classifiers’ predictions. We use random forests (RFs) and k-nearest neighbour (k-NN). Random forests consist of an ensemble of decision trees [33]. A decision tree is a classifier consisting of recursive binary questions with each final node assigning a probability of the training sample belonging to each possible class. The individual trees are then combined and weighted in order to produce a classification. Whilst the single decision trees are each weak classifiers when combined they produce accurate results. k-NN is a non-parametric method which predicts the testing samples based on the votes of k nearest training samples in the feature space [34]. Each of these was trained and tested independently using the feature vector generated in the previous step as input. Different parameters within the models were tested to produce and optimal classification accuracy.

#### 2.3.3. Neural Networks

NNs are a set of machine learning models in which the most pertinent discriminatory features are automatically learnt from training data. These are combined in the same pipeline as the classifier and remove the need for handcrafted feature extraction approaches. The shortcomings of using handcrafted techniques include the necessity of making predictions on the data about which features will be the most discriminatory to use; neural networks are able to learn this information rather than making assumptions.

They combine feature learning and classification into a single pipeline. They consist of multiple connected, weighted layers in which different levels of features are extracted. The non-linearity of functions between layers allows basic features be combined to produce complicated patterns. Different types of layers are suitable for extracting different features. In convolutional layers, a bank of locally receptive filters convolve across the input data. These are then able to identify local features by examining the relationship between neighbouring data points of the input results in the formation of spatial features. At the backward pass stage, these filters are automatically optimised via backpropagating the prediction error of the forward pass. Fully connected layers are also included in the network; in these, all nodes from one layer are connected to all nodes in the next with weightings updated in the same way. This allows pertinent localised features to be more easily identified. Recurrent networks are particularly suited to dealing with temporal data and are commonly used for problems such as natural language translation, weather prediction and understanding video clips. They contain a loop, such that each cell takes in the output of the previous cell as well as a sequential input. This allows information to be propagated through the steps of the network. Long short-term memory networks (LSTMs) are a form of recurrent network which are able to regulate information efficiently and appropriatley decide which information is unimportant and which is useful. A final classification layer is needed to predict the probability of the input belonging to each of the possible classes. The class with the highest probability becomes the predicted word. To feed into this layer a fully connected layer with the same number of nodes as there are classes is required. During training, the prediction are compared to the ground truth, from which loss can be calculated and backpropagated through the network to update all weightings.

A major drawback of neural networks is that they require a large amount of data for training. There are a number of solutions to this which we use: having a shallower network with few layers, having fewer nodes at each layer, introducing dropout layers and rectified linear units (ReLU) layers. ReLU produces an output between [0, *∞*) with all negative input mapped to 0. They are used with the activation functions for regularisation and reduce computation time. A dropout layer randomly ignores a set of neurons in a fully connected layer during each training batch which helps to reduce overfitting [35]. This means that these nodes are not considered during a single forward and backward pass. At each stage, individual nodes are either dropped with a probability of *p* or kept with a probability of 1−p, leaving a reduced network. This means that for each pass of the same input, different nodes are dropped preventing the architecture getting so tuned to the training set.

We develop a neural network containing a convolutional layer and an LSTM layers, as shown in Figure 2. This combination allows for different temporal features to be determined. Dropout layers and a ReLU layer are included to reduce overfitting. A softmax layer is used as the classifier to calculate the error across the network.

## 3. Results and Discussion

### 3.1. Device Characterisation

Characterisation of the physical properties of the device under strain is shown in Figure 3. The Keithley 2400 Source Meter Unit with the associated KickStart software was used for resistance readings. The Deben Microtest Tensile Stage was used for applied strain measurements. The device showed a broad sensing range with a linear response of up to 15% elongation which is sufficient for the intended purpose. For repeated strain–release cycles, a repeatable pattern of resistance changes is noted. However, there is some drop off in initial cycle phases although this slows as number of cycles increase. Some works which use polymer encapsulation demonstrate higher durability over repeated strain cycles [10,11]. However, other devices made by dyeing fabric in graphene also demonstrate a similar decrease in resistance as applied cycles increase [14]. This suggest encapsulation can be useful for protecting the device from damage.

Figure 4a–d shows scanning electron microscope (SEM) images of the graphene-coated lycra device at different magnifications for either unused devices or one which has been through multiple applied strain cycles. (a) and (b) indicate that the graphene sheets connect many fibres in the same wire. (c) and (d) show in closer detail the changes caused to the graphene after use of the device. In (c), a rough but continuous morphology is seen. In (d), the graphene flakes appear less dense following stretching of the fibres. This suggests there has been some exfoliation and that the morphology of the flakes do not return to their initial state. This would explain why there is a change in resistance after multiple strain cycles.

### 3.2. Dataset and Observations

The two datasets consisted of 15 different words and, 4 different phonations respectively, each with 20 repetitions. These were recorded at 1000 Hz.

In Figure 5 we demonstrate some of the recordings for a selection of phonations. Shown are the words “me”, “none”, “up” and “us” and the movements yawn and nod. As can be seen there is some visible similarities between the different recordings of the same word. Similar resistance signal shapes can be seen for the repetitions of the same phonation. This is more obvious for some than others. For example, the *yawn* seems to have a consistent gradual increase in signal before a sharp decrease. “none” shows a similar pattern of a very sharp increase in resistance followed by a decreased plateau. Some examples show consistency between a number of repetitions but with some intraclass variability. These inconsistencies are likely brought about by movements of the neck which produce sharp spikes of noise and by an imperfectly fitting device. There is, however, some element of similarity between the repetitions which makes it possible to group them into the same class.

Similarly, we compare the inter-class signals. There is some evidence for distinct features in each class. This is most obvious in the movements dataset where there seems to be greater differentials between each movement. These movements are more extreme than the subtle word differences and so the resultant resistance signals should be more differentiable. To some extent, visual differences can also be seen between each of the different words. However, these are not always clear. For example, there are similarities between “none” and “up”. This is seen across the whole dataset with many of the words having similar signal patterns. This makes the differentiation between groups difficult. High resolution data are required for these to detect the subtle differences between similar signals.

In other works there has been a demonstration of either movements or words which have shown high reproducibility and repeatability with repetitions appearing visually very similar. However, in these instances, a maximum of three repetitions are shown [10,12,13,14]. This suggests that such designs may be more receptive to the small throat movements, and the utilisation of such designs could lead to more accurate classifications. However, these are more complex designs, many involving polymer encapsulation, and are less wearable than our design. Further exploration of the fabrication approach of a wearable and receptive device could improve prediction accuracies whilst remaining viable as a long-term wearable device.

### 3.3. Classification

The layer architecture of the used neural network is shown in Figure 2. All neural networks consist of a number of parameters and hyper-parameters which have to be chosen to ensure optimal performance of the architecture. It can be difficult to select all parameters correctly with the optimal settings changing depending on architecture, data and goal. The parameters for the used model were decided through testing. For the neural network, the convolutional layer contained 120 nodes, the LSTM layer contained 80 nodes, the dropout layers had a dropout rate of 0.6. Standard Gradient Descent was used to optimise weightings and an initial learning rate of $10^{-4}$ was used. The model was trained for 100 epochs as training loss reached a minimum during this time. The random forest contained 100 classification trees and was trained for 20 cycles. A k value of 5 was used for k-NN.

Leave-one-out cross-validation was used. This approach consists of splitting the data into a training set and a testing set. The testing set includes a single member of each class while the training set contains the other 19 samples from each class. This is repeated with each sample of a given class being included in the training set in turn leading to 20 models being trained and tested. This whole process was repeated five times, giving 100 training and testing runs in total. The reported results were the mean across these runs.

Table 1 shows a comparison of the accuracies for each of the three classification methods on the noises and movements datasets. It can be seen that the NN outperforms the RF and k-for both datasets. This is unexpected on a small dataset, as NNs normally require large amounts of data to be able to accurately pertain the most important features. One of the challenges of these data was knowing which the most important elements were when performing handcrafted feature extraction. We concentrated on time and frequency domain features which had been shown to work for other 1-dimensional data classification applications. The lower classification accuracies produced from these handcrafted features suggest they were not the optimal choice. A greater exploration of which features are most important for our signal type would be required to further understand the discriminatory factors between interclass signals from these resistance readings which could lead to higher accuracies.

Table 2 and Table 3 show the confusion matrices for the movement and word datasets, respectively, for the NN—the best performing of the classification approaches. Similar trends were seen for all three classification methods for the words which were most and least likely to be accurately predicted.

The noise dataset produced a mean prediction accuracy of 54.6% across the 15 classes. There was a wide range in the accuracy of individual words. This suggests variation in the uniqueness of throat movements required for different words. The words “me” and “up” performed well, with prediction accuracies of 90.0% and 95.5%, respectively. Examples of these signals are shown in Figure 5. Visually, we are able to see a consistency between samples in the same group and uniqueness in signal shape from other classes. Conversely, some words such as “give” performed poorly with a 2.9% prediction accuracy. This is less than random chance and suggests that unique features from it are difficult to discover. This range in classification implies that different words do cause different throat vibration patterns which can be distinguished between, and there is merit in this approach to silent speech recognition.

For the movements dataset, an overall prediction accuracy of 84.5% was achieved on the testing data. These movements were larger and more distinct than the words, and so a higher accuracy was expected. All four movements were predicted with an accuracy of over 70% with *yawn* performing particularly well with 94.8% correct prediction rate. The action of yawning is longer than the other movements and so produces distinct temporal features not observed in the others. A demonstration of an ability to discriminate between these actions is shown, though a more diverse dataset would test the robustness of the models.

The neural networks are able to learn which features within the temporal resistance data are most important and are most likely to identify the correct class. There is still an element of overfitting from all the models, as training accuracies of over 95% were produced for all models on each of the datasets. Overfitting is caused by the model fitting too closely to the training data and losing its ability to generalise to unseen data. It overvalues the importance of features which are unique to individual instances but would not occur in unseen data. Overfitting was mitigated to an extent through the simplification of the model designs, but the size of the training set is the main contributor to this problem. This is particularly true when there are a number of potential classes. This problem can only properly be overcome by developing a much larger dataset.

This is the first work to quantitatively assess the performance of strain sensors by classification for speech recognition. These results demonstrate the plausibility of using machine learning to automatically classify resistance readings from wearable strain gauges into predicted speech. It provides a proof-of-concept with further improvements in device sensitivity and larger datasets likely to lead to improved accuracies.

## 4. Conclusions

The aim of this work was to produce a wearable graphene-based strain sensor prototype which could detect small muscle movements and vocal vibrations and convert them into words. This involved fabricating a suitable device, developing a dataset of spoken words and movements, and designing, training and testing machine learning algorithms to classify measured signals into words. A proof-of-concept approach has been demonstrated with a prediction accuracy of 55% being achieved on a dataset of 15 different words and an 85% accuracy on a dataset of four different words.

The chosen method for the strain gauge was to print graphene ink onto lycra fabric. A previous study demonstrated that dyed polyester fabric with rGO could produce a suitably sensitive sensor [14]. This approach offers a number of benefits with the produced sensor being highly wearable and flexible allowing high contact with the human body. The fabrication process was also much simpler and quicker than other techniques which involved polymer encapsulation [12,13]. Our process involved depositing graphene ink onto the fabric, unlike [14] who dyed the fabric. It was, however, not exact and relied on human judgement to decide when a suitable thickness of material had been applied. This makes it difficult to fabricate multiple, consistent sensors. It was found that if the graphene is not consistently applied the sensor produces unpredictable changes in resistance when strain is applied. The drawbacks of our sensor include a relatively high resistance and an unpredictable nature of the fabric. Lycra is easy to deform which affects durability and the irregular nature of the weaved structure affects repeatability. A more sensitive, reliable and durable sensor may produce signals which are better for developing accurate classification algorithms.

We developed a word set of 15 words and a movement set of four movements. Each of these contained 20 repetitions. The words were selected so as to be phonetically balanced. The repetitions allowed classifiers to be trained and tested which had not been done in other works. However, this dataset could still have been much larger as accurate machine learning approaches require substantial amounts of training data. A dataset with a diverse set of test data allows more robust and useful models to be trained. Additionally, a greatly expanded range of words would be required before it moved to a conceivable useable product.

Two approaches to classification were used. Firstly, we used handcrafted features fed into RFs and k-NN classifers, and secondly, we tested learning features automatically using NNs. We were able to produce a set of models which were able to produce signal to word classifications at a much higher accuracy than random chance. This demonstrates there to be underlying features in the acquired signals which can be used to discriminate between different words. The highest accuracies of 55% and 85%, for the word and movement datasets, respectively, are demonstrative of this, but are below what would be required for a useable system. Restrictions on the accuracy of the algorithms is due to the limitations of the quality of the recorded signal and the size of the training set, as discussed above. With larger datasets more complex and deeper NN models could be produced, following a similar design. This would be better able to extract pertinent features from the data and would lead to better prediction accuracies on test data as overfitting is reduced. 

## Figures and Tables

**Figure 1 sensors-22-00299-f001:**
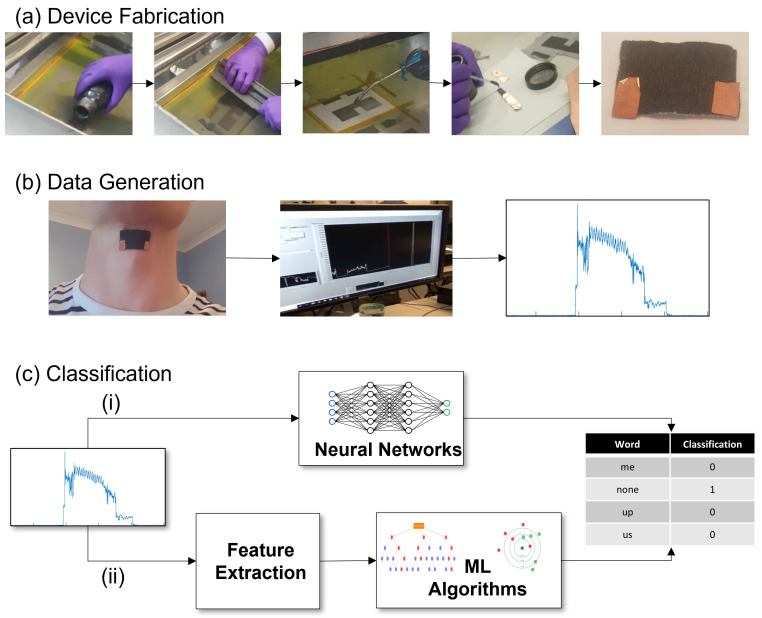
Flowchart of the proposed method. (**a**) Device Fabrication—Graphene ink is screen printed onto lycra fabric ensuring a full coverage. The device is cut to size and electrodes area attached using silver paste. (**b**) Data Generation—The strain gauge sensor is worn on the user’s throat of the subject. The electrodes area attached to an ohmmeter to make resistance recordings across the device. (**c**) Classification—The data are labelled and split into training and testing sets. For classification either, (**i**) The raw signal data is passed to a convolutional-recurrent artificial neural network, as shown in Figure 2, which learns the pertinent features during its training phase and makes predictions on unseen test data, or (**ii**) Handcrafted features are extracted and random forests or k-nearest neighbour classifiers are used for the training and testing of the data.

**Figure 2 sensors-22-00299-f002:**
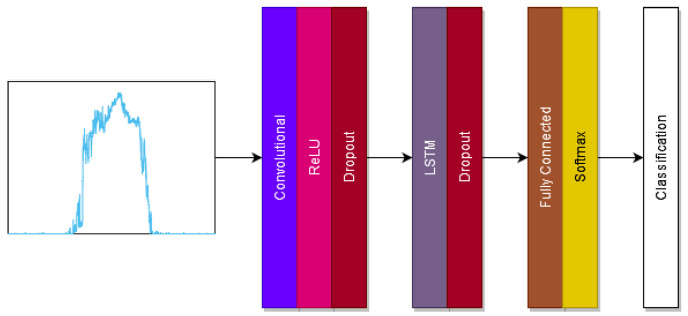
Neural network used for signal to phonation classification. It consists of a convolutional layer, a ReLU layer, an LSTM layer, two dropout layers, a fully connected layer and uses a softmax classifier.

**Figure 3 sensors-22-00299-f003:**
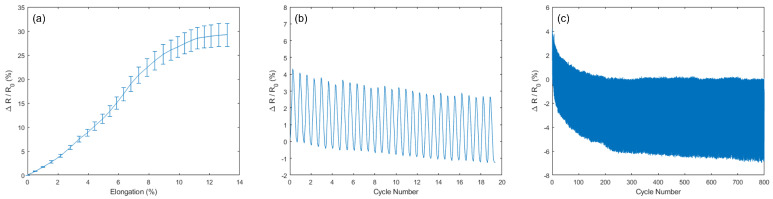
Performance of graphene strain gauge. (**a**) Relative resistance changes with elongation of the device (**b**,**c**) Relative resistance from multiple strain cycles at 1% strain.

**Figure 4 sensors-22-00299-f004:**
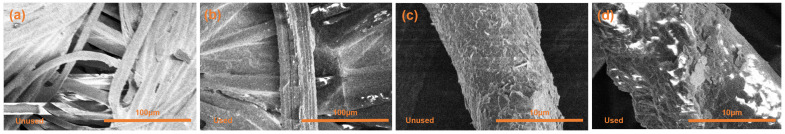
SEM images of graphene strain gauge for: (**a**,**c**) unused device, (**b**,**d**) used device.

**Figure 5 sensors-22-00299-f005:**
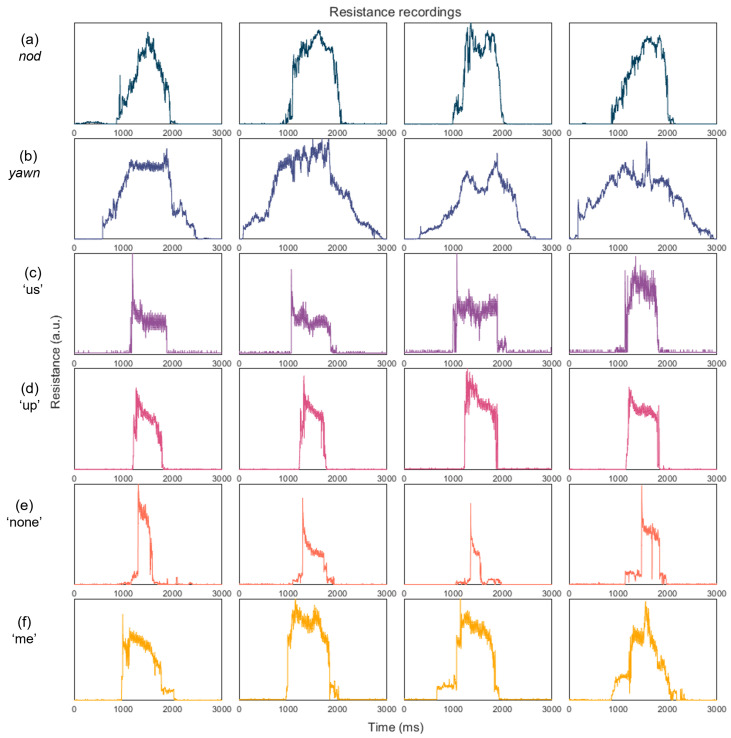
Examples of resistance readings of (**a**) *nod*, (**b**) *yawn*, (**c**) ’us’, (**d**) ’up’, (**e**) ’none’, (**f**) ’me’.

**Table 1 sensors-22-00299-t001:** Dataset prediction accuracies (%).

	RF	k-NN	NN
**Words**	50	46	55
**Noises**	82	81	85

**Table 2 sensors-22-00299-t002:** Confusion matrices of noises predictions for NN (%).

	Cough	Nod	Swallow	Yawn	Avg.
**cough**	**88.5**	14.8	11.6	2.1	**84.5**
**nod**	5.5	**72.8**	1.7	0.3	
**swallow**	5.5	10.7	**81.8**	2.8	
**yawn**	0.5	1.6	5	**94.8**	

**Table 3 sensors-22-00299-t003:** Confusion matrices of noises predictions for NN (%).

	As	Dad	Felt	Give	Hum	It	Low	Me	None	Or	Poor	There	True	Up	Us	Avg.
**as**	**66.6**	2.4	0	5.7	2.9	2.4	0	0	4.8	1.6	3.4	0	3.4	0.7	0	**54.6**
**dad**	21.9	**86.8**	25.2	13.3	11.6	2.4	0	0.7	6.4	12.5	0	0	0	0	2.4	
**felt**	1.20	0.6	**17.4**	11.4	1.5	6.1	5.3	1.4	3.2	1.6	3.4	3.3	1.7	0	3.6	
**give**	0	0	1.8	**2.9**	0	0	0	0	0	1.6	0	2.2	0	0	0	
**hum**	5.8	0.6	16.2	7.6	**49.2**	0	3.9	3.6	3.2	0	8.5	3.3	0	0	6.1	
**it**	1.2	0.6	3.6	1.9	1.5	**63.7**	2.6	0.7	8	3.1	5.1	0	5.1	1.4	1.2	
**low**	1.2	3	1.8	13.3	0	2.4	**57.9**	0	4.8	11	3.4	3.3	3.4	0	1.2	
**me**	1.2	0.6	9	1.9	10.2	2.4	0	**90**	4.8	1.6	6.8	2.2	3.4	0.2	4.8	
**none**	1.2	0.6	3.6	11.4	2.9	2.4	7.9	0	**37.6**	1.6	8.5	3.3	6.8	0	3.6	
**or**	0	0.6	1.8	7.6	2.9	3.6	10.5	0	6.4	**40.5**	8.5	4.4	5.1	0.2	2.4	
**poor**	0	0	1.8	0	0	2.4	0	0	0	0	**28.9**	1.1	3.4	0	0	
**there**	0	0	7.2	9.5	4.4	1.2	3.9	0.7	4.8	9.4	16.9	**69.5**	11.8	0.7	2.4	
**true**	0	0.6	1.8	1.9	1.5	0	1.3	0	4.8	3.1	5.1	2.2	**28.9**	0.7	1.2	
**up**	0	3	5.4	5.7	2.9	3.6	5.3	0	6.4	3.1	0	2.2	16.9	**95.5**	7.3	
**us**	0	0.6	3.6	5.7	8.7	7.3	1.3	2.8	4.8	9.4	1.7	3.3	10.2	0.5	**63.7**	

## Data Availability

The data presented in this study are available on request from the corresponding author.

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
