# Peer review of "Machine Learning Methods for Automatic Silent Speech Recognition Using a Wearable Graphene Strain Gauge Sensor"

_sensors, 2021, doi:10.3390/s22010299_

Round 1

Reviewer 1 Report

Given the rise in wearable sensors and machine learning methods, the topic studied in this paper is interesting, but I have the following comments.

Comments:

  1. The author explained the background of strain gauge sensors for silent speech recognition well, but the paper does not explain the machine learning approach for this problem.
  2. The author should clearly state this paper’s novelty and main contributions in the introduction.
  3. Figure 1(c) contains an error of illustration between (i) and (ii) that is a conflict with its figure description.
  4. In the neural network that the author developed (CNN-LSTM), the model's parameters such as sizes of filters or feature maps must be given with reason, i.e., empirical results or literature. The justification for the requirement of the layers has to be mentioned.
  5. The results in section 3.3 must be discussed and compared with state-of-the-art methods.

Author Response

Thank you for your feedback. As detailed below, we have adapted the paper according to your comments:

  1. Explanations of other machine learning approaches applied to temporal problems have been assessed in the introduction. These are used to justify our approach.
  2. The novelties in this paper include the development of a structured dataset, the use of machine learning approaches for classification, and the first recorded approach for automatic classification of resistance readings. These have been more clearly stated in the Introduction.
  3. This mistake has been corrected and the figure is now correct.
  4. Parameters were developed through testing. This is mentioned in the paper.
  5. These have now been explained in more detail. Comparisons to other literature has been made for the characterisation of the device and for observations of the resistance readings. However, as ours is the first approach to providing a quantitative evaluation predicting speech, there is no comparison possible for the classification accuracies.

Hope you enjoy the Christmas holidays.

Many thanks,

Dafydd

Reviewer 2 Report

This paper presents a very interesting application of the graphene strain gauge sensor for silence speech recognition. The authors manufactured a graphene strain gauge sensor, which is applied to collect data corresponding to 15 unique words and four movements. Different machine learning methodologies have been applied for the recognition of different words/movements, with sufficient evaluation studies being provided.

Author Response

Thank you for your feedback. 

Reviewer 3 Report

  1. What device and software were used in figure 1. b?
  2. In figure 1. c: what type of neural network (layers numbers and neurons allocated) were used?
  3. R302: what type of graphene was used? particles size?
  4. Figure 4: The authors should add a comment about differences between used and unused devices? Seams to be exfoliation of the graphene on the textile yarns.
  5. For figures 4. c and 4. d it would be interesting to add also an EDAX if the authors performed this investigation. Maybe the elemental analysis offers more conclude information about the decreased percent of graphene (C).
  6. Figure 5: what represents the representations in different colors? The titles for axis Ox and Oy are missing.
  7. What type of validation was used?
  8. In the part of the discussion, the authors should add a comparison with other approaches from the scientific literature.

Author Response

Thank you for your feedback. As detailed below, we have adapted the paper according to your comments:

  1. The Keithley 2400 Source Meter Unit with the associated KickStart software was used for resistance readings. The Deben Microtest Tensile Stage was used for applied strain measurements. These details have been added to the paper.
  2. This has been clarified.
  3. The graphene ink was produced via liquid phase exfoliation using the method proposed by Marcano, as cited in the paper. This produces flakes in the range 50-150nm with thicknesses of 1-3nm. More details have been added to the paper.
  4. Additional comments have been made about the differences between the used and unused devices.
  5. This would be interesting to explore, however, unfortunately, due to the short turnaround and timing of the resubmission process we will be unable to include this analysis in this paper.
  6. Each row represents examples from different categories. The x axis for all is Time (ms) and y axis is Resistance (a.u.). These have been updated to be more clear.
  7. For classification leave-one-out-cross validation was used. This has been explained in Section 3.3.
  8. These have now been explained in more detail. Comparisons to other literature has been made for the characterisation of the device and for observations of the resistance readings. However, as ours is the first approach to providing a quantitative evaluation predicting speech, there is no comparison possible for the classification accuracies.

Hope you enjoy the Christmas holidays.

Many thanks,

Dafydd